# Presence of Cutaneous Complement Deposition Distinguishes between Immunological and Histological Features of Bullous Pemphigoid—Insights from a Retrospective Cohort Study

**DOI:** 10.3390/jcm9123928

**Published:** 2020-12-03

**Authors:** Sascha Ständer, Maike M. Holtsche, Enno Schmidt, Christoph M. Hammers, Detlef Zillikens, Ralf J. Ludwig, Khalaf Kridin

**Affiliations:** 1Department of Dermatology, University of Lübeck, 23562 Lübeck, Germany; Sascha.Staender@uksh.de (S.S.); MaikeMarleen.Holtsche@uksh.de (M.M.H.); enno.schmidt@uksh.de (E.S.); detlef.zillikens@uksh.de (D.Z.); Ralf.Ludwig@uksh.de (R.J.L.); 2Lübeck Institute of Experimental Dermatology, University of Lübeck, 23562 Lübeck, Germany; Christoph.Hammers@uksh.de

**Keywords:** BP, bullous pemphigoid, complement, direct immunofluorescence, immunological, morphological

## Abstract

The practical implications of complement deposition in direct immunofluorescence (DIF) microscopy and its influence on the disease phenotype are poorly understood. We aimed to investigate whether the presence of complement deposition in DIF microscopy gives rise to differences in the morphological, immunological, and histological characteristics of patients with BP (bullous pemphigoid). We performed a retrospective study encompassing patients with BP in a specialized tertiary referral center. Logistic regression model was utilized to identify variables independently associated with complement deposition. The study included 233 patients with BP, of whom 196 (84.1%) demonstrated linear C3 deposition along the dermal-epidermal junction (DEJ) in DIF analysis. BP patients with C3 deposition had higher mean (SD) levels (645.2 (1418.5) vs. 172.5 (243.9) U/mL; *p* < 0.001) and seropositivity rate (86.3% vs.64.9%; *p* = 0.002) of anti-BP180 NC16A and less prevalent neutrophilic infiltrate in lesional skin specimens (29.8% vs. 52.4%; *p* = 0.041). C3 deposition was found positively associated with the detection of anti-BP180 NC16A autoantibodies (OR, 4.25; 95% CI, 1.38–13.05) and inversely associated with the presence of neutrophils in lesional skin (OR, 3.03; 95% CI, 1.09–8.33). To conclude, complement deposition influences the immunological and histological features of BP. These findings are in line with experimental data describing the pathogenic role of complement in BP.

## 1. Introduction

Bullous pemphigoid (BP) is a prototypical organ-specific subepidermal autoimmune bullous dermatosis, typified by autoantibodies targeting the hemidesmosomal proteins BP180 and/or BP230, which are expressed by basal keratinocytes abutting the epithelial dermal-epidermal junction (DEJ). Most autoantibodies are directed against the NC16A domain of BP180 [1]. Clinically, most patients manifest with tense bullae often in conjunction with pruritic urticarial plaques [2]. BP is associated with a life-threatening potential and often necessitates prolonged administration of topical or systemic corticosteroids, as well as adjuvant immunosuppressive agents [3,4,5].

Deposits of complement components, of which C3 deposition is of high diagnostic significance, are typically observed in patients with BP and gestational pemphigoid [6]. Beyond the diagnostic importance of complement factors, it is thought that complement activation plays an important role in the pathogenesis of BP [7]. While early in the neonatal mouse model of BP activation of complement appeared as a prerequisite for blister induction by anti-BP180 IgG [8,9,10], subsequent in vitro experiments and studies in a humanized and an adult mouse model of BP challenged this conclusion by demonstrating complement-independent mechanisms of subepidermal blister formation [11,12,13,14,15].

Although the role of complement has been extensively investigated in experimental mouse models and human keratinocyte cultures, the practical implications of complement deposition and its potential impact on the clinical and immunological features of patients with BP are poorly understood. A recent study revealed a correlation between the complement-activation capacity of autoantibodies, measured by complement binding test, and the severity scores of BP [16]. Romeijn et al. [17] found that complement deposition in direct immunofluorescence (DIF) analysis was associated with higher levels and seropositivity of pathogenic autoantibodies. 

The aim of the current study was to investigate whether the presence of complement deposition in DIF microscopy is associated with distinct morphological, immunological, and histological characteristics in BP patients. We additionally sought to identify factors independently associated with complement deposition and to reveal differences between patients with isolated complement deposition relative to those with deposition of complement alongside immunoglobulins in DIF microscopy. 

## 2. Methods

### 2.1. Study Population and Definition of Patients 

We performed a retrospective cohort study encompassing patients diagnosed with BP between 1 January 2009, and 28 February 2020, in a referral center of autoimmune bullous diseases at the Department of Dermatology, University of Lűbeck, Lűbeck, Germany. The current study was approved by the institutional review board (20-110A).

The diagnosis of BP was based on the following criteria: (i) suggestive clinical presentation; (ii) linear deposits of immunoreactants along the DEJ by DIF microscopy of a perilesional skin biopsy; (iii) detection of circulating autoantibodies binding to the epidermal side of 1 mL salt-split normal human skin by indirect immunofluorescence (IIF) microscopy and/or the presence of circulating IgG autoantibodies against BP180 NC16A and/or BP230, as identified by enzyme-linked immunosorbent assay (ELISA) [18,19]. 

Patients lacking detailed documentation of the positive immunoreactants in DIF or patients in whom DIF was biopsied under immunosuppressive therapy were excluded from the current study. Appendix A summarizes the selection of the current study population.

### 2.2. Definition of Covariates

The severity of BP was evaluated based on the Bullous Pemphigoid Disease Area Index (BPDAI) [20]. This score had been documented, including four subcomponents (cutaneous erosion/blister activity, cutaneous urticaria/erythema activity, damage, and pruritus). Since the aforementioned scoring system was introduced only in 2012, BPDAI scores were available for only 101 out of 233 (43.3%) patients. 

Previously published protocols were followed for DIF and IIF of patient sera on 1 M salt-split normal human skin and monkey esophagus [18]. Fluorescein-conjugated goat polyclonal IgG to human complement C3 was utilized as a reagent in immunofluorescence assay and tissue staining. The total protein was measured using the Biuret procedure with bovine albumin as the standard. F/P ratio was calculated using absorbance at 491.5 nm. The levels of circulating anti-BP180 NC16A and anti-BP230 autoantibodies were measured utilizing commercial ELISA systems (Euroimmun, Lűbeck, Germany). Seropositivity was defined based on the cutoff values proposed by the manufacturer (i.e., 20 U/mL). Based on the histopathological description, lesional skin biopsies were divided into “eosinophil-dominant”, “neutrophil-dominant”, and lymphocyte-dominant. Circulating eosinophil counts and C-reactive protein levels were measured prior to the administration of any systemic management. Peripheral eosinophilia refers to an absolute count of ≥500 eosinophils/µL in peripheral blood [21]. 

### 2.3. Statistical Analysis

Baseline characteristics were described by means and standard deviations (SD)s for continuous variables, whilst categorical values were described by percentages. The comparison of clinical, immunological, and immunopathological variables between subgroups was performed using the Chi-square test and *t*-test for categorical and continuous variables, respectively.

To identify factors associated with C3 deposition, a logistic regression model was used to calculate odds ratios (ORs) and 95% confidence intervals (CIs). Multivariate analyses were then performed using stepwise logistic regressions, with enter and removal limits set at *p* < 0.200. SPSS software, version 25 (SPSS, IBM Corp, Armonk, NY, USA), was utilized to conduct all statistical analyses.

## 3. Results

### 3.1. Demographic Characteristics of the Study Population

The study cohort included 233 patients with BP, of whom 132 (65.7%) were females and 101 (43.3%) males. The mean age (SD) at diagnosis was 79.2 (9.7) years, and the median (range) age was 80.7 (49.6–98.2) years.

### 3.2. The Main Results of Direct Immunofluorescence Analysis

Overall, 196 (84.1%) patients demonstrated linear C3 deposition along the DEJ in DIF analysis. C3 represented the most frequently detected immunoreactant by DIF analysis and was followed by IgG (*n* = 188; 80.7%), IgA (*n* = 23; 9.9%), and IgM (*n* = 7; 3.0%). While 73 (31.3%) patients had a deposition of a single immunoreactant, 139 (59.7%) and 21 (9.0%) patients exhibited a simultaneous deposition of two and three immunoreactants, respectively. The most frequent patterns of immunoreactant deposition was the co-occurrence of IgG and C3 (*n* = 132; 56.7%), followed by isolated C3 (*n* = 41; 17.6%) and isolated IgG (*n* = 31; 13.3%) deposition.

### 3.3. Features of Patients with C3 Deposition

We then addressed the different characteristics of patients with (*n* = 196) and without (*n* = 37) C3 deposition in DIF microscopy analysis. Table 1 demonstrates that the two subgroups were comparable with regard to the demographic and morphological variables, although patients with C3 deposition tended to present with less frequent acral (82.1% vs. 94.6%; *p* = 0.057) and more frequent mucosal (12.8% vs. 2.7%; *p* = 0.075) involvement.

BP patients with C3 deposition had higher mean (SD) levels (645.2 (1418.5) vs. 172.5 (243.9) U/mL; *p* < 0.001; Figure 1) and seropositivity rate (86.3% vs.64.9%; *p* = 0.002) of anti-BP180 NC16A IgG. However, the levels and seropositivity of anti-BP230 IgG autoantibodies, as well as the detection rate of anti-DEJ autoantibodies by IIF were comparable among patients with and without C3 deposition (Table 2).

When the deposition of additional immunoreactants in DIF microscopy was examined, patients with C3 deposition had less frequent concomitant deposition of IgG (77.6% vs. 97.3%; *p* = 0.005). Expectedly, concomitant deposition of two (68.4% vs. 13.5%; *p* < 0.001) and three (10.7% vs. 0.0%; *p* = 0.037) immunoreactants was more frequent among patients with C3 deposition.

On the histological level, the presence of neutrophilic infiltrate was significantly less prevalent in lesional skin specimens of patients with C3 deposition (29.8% vs. 52.4%; *p* = 0.041). While the presence of lymphocytes was also less apparent among patients with C3 deposition (64.1% vs. 85.7%), it fell marginally short of significance (*p* = 0.051). No other histological differences were revealed between the subgroups (Table 2). Similarly, the levels of peripheral eosinophil count and C-reactive protein were not statistically different (Table 2).

### 3.4. Factors Associated with C3 Deposition

In univariate analysis, C3 deposition was significantly and positively associated with seropositivity of anti-BP180 NC16A IgG (OR, 3.42; 95% CI, 1.55–7.54) and inversely associated with IgG deposition in DIF (OR 0.10; 95% CI, 0.01–0.72) and the presence of neutrophils in lesional skin specimens (OR, 0.39; 95% CI, 0.15–0.98). In multivariate logistic regression analysis, C3 deposition emerged as an independent predictor of anti-BP180 NC16A IgG seropositivity at baseline (OR, 4.25; 95% CI, 1.38–13.05) as well as lack of neutrophils in histological specimens (OR, 0.33; 95% CI, 0.12–0.91; Table 3).

### 3.5. Features of Patients with Isolated C3 Deposition

The last endpoint of the current study was to compare the features of patients with isolated C3 deposition (*n* = 41) relative to patients with C3 deposition in conjunction with other immunoreactants (*n* = 155; Table 4). Patients with isolated C3 deposition were characterized by lower levels of anti-BP180 NC16A IgG (273.6 [388.3] vs. 738.1 [1561.5]; *p* = 0.001) and anti-BP230 IgG (16.7 [20.0] vs. 94.7 [175.4]; *p* = 0.007) and lower detection rate of anti-DEJ antibodies by IIF on human salt-split skin (85.0% vs. 96.1%; *p* = 0.010).

Of note, patients with C3 deposition showed lower prevalence of eosinophil-dominant (63.2% vs. 83.7%; *p* = 0.041) and higher prevalence of lymphocyte-dominant (31.6% vs. 10.9%; *p* = 0.019) inflammatory cell infiltration in lesional skin biopsies. Figure 2 graphically summarizes the main findings of the current study.

## 4. Discussion

The current study demonstrated that complement deposition occurred among 84.1% of patients with BP, representing the leading immunoreactant in DIF microscopy. Complement seems to influence the immunological and histopathological features of BP as patients with C3 deposition presented with higher levels and higher detection rate of BP180 NC16A autoantibodies as well as with less frequent neutrophilic infiltrate in lesional skin biopsies. Detection of BP180 NC16A autoantibodies and the absence of neutrophils were found to be independently associated with complement deposition in multivariable analysis. Compared to patients with deposition of C3 alongside other immunoreactants, those with isolated C3 deposition had lower levels of anti-BP180 and anti-BP230 autoantibodies, lower frequency of eosinophil-dominant, and higher frequency of lymphocyte-dominant inflammatory infiltrate in lesional skin biopsies.

### 4.1. The Role of Complement in the Pathogenesis of BP

The complement system was long considered as a crucial player in the pathogenesis of BP. Comprehension of the pathogenic role of the complement in the pathogenesis of BP was facilitated by the neonatal mouse model based on the passive transfer of a rabbit antibody against murine BP180 NC14A, a homolog of the human immunodominant BP180 NC16A domain [8]. Utilizing this model, Liu et al. [8] demonstrated that C5- or C4-deficient mice were protected against the pathogenic effect of anti-murine BP180 IgG, and that depletion of serum complement by cobra venom interfered with the development of the clinical phenotype of BP. Additionally, the transfer of F(ab′)2-fragments derived from the anti-murine BP180 antibody did not manage to induce a BP phenotype, signifying the pivotal effect of IgG-Fc portion that bears the C1q binding region and is crucial for activation of the classical pathway of the complement system [8]. Correspondingly, C4-deficient mice, as well as wildtype mice treated with a function blocking anti-murine C1q antibody, were proved resistant to develop BP [7]. The alternative pathway of the complement system was also implicated in the pathogenesis of BP when factor B-deficient mice were found to develop delayed and less intense blisters [9,10].

Some of the aforementioned observations were reproduced in a humanized mouse introducing the human BP180 cDNA transgene into BP180-null mice [22]. Liu et al. [23] revealed that complement depletion by cobra venom factor protected against the development of BP in their humanized BP-model model replacing the murine NC15A with its human homologue NC16A. Consistently, the transfer of anti-human BP180 NC16A IgG1 mutated at the C1q binding site did not produce BP phenotype [24]. In a similar experimental model of the antibody transfer-induced mouse model of epidermolysis bullosa acquisita (EBA), the C5a/C5aR1 axis is a main drive of skin pathology [25,26]. Based on these findings, antibodies targeting the classical complement activation, as well as the C5a/C5aR1 axis are currently developed [27,28,29].

Subsequently, several lines of evidence have accumulated to suggest the existence of complement-independent mechanisms mediating blister formation in BP. Autoantibodies from BP patients were found to directly deplete cultured human keratinocytes of BP180 [13]. Furthermore, the internalization of the IgG–BP180 complex through endocytosis has been observed [14]. Using a model of adult mice, Karsten et al. [15] had recently demonstrated that skin lesions decreased by only 50% when C5-deficient mice were injected with anti-BP180 IgG, opposing the absence total skin lesions observed in the C5-deficient non-humanized BP-model [8]. Passive transfer of autoantibodies from patients with BP induced blister formation in neonatal C3-deficient BP180-humanized mice [12]. Additionally, F(ab′)2 fragments of anti-BP180 IgG antibodies from BP patients or rabbit IgG against humanized NC16A induced partial skin fragility in neonatal humanized mice [11].

In addition, generation of C3 in the skin may be important beyond autoantibody-induced tissue damage. In mouse models of cutaneous herpes simplex virus (HSV) infection or delayed type hypersensitivity C3^−/−^ mice did have an impaired antibody response towards HSV or the allergen applied on the skin [30,31]. Collectively, this indicates that C3 promotes antibody production. Our findings of an increased seropositivity and levels of anti-BP 180 NC16A autoantibodies in BP patients with C3 deposits in the skin supports this notion.

On the clinical level, two patients lacking complement deposition in DIF presented with BP in which IgG4 antibodies, typified by limited complement-fixing ability, were the dominant antibody [32]. Moreover, IgG4 antibodies were reported to predominate in some patients with BP, particularly those without deposition of complement [33], and BP developed in a patient with C4-defecicney [34]. Taken together, these experimental and clinical findings denote that complement-independent mechanisms may play a role in the pathogenesis of BP and be of phenotypical relevance.

### 4.2. Interpretation of the Main Study Findings

The current study indicated that patients with C3 deposition presented with significantly higher levels of anti-BP180 IgG and that C3 deposition in DIF was independently associated with seropositivity of anti-BP180 autoantibodies. These findings accord with those reported by Romeijn et al. [17] and probably reflect the imperative role of the complement system in the pathogenesis of BP. As the levels of anti-BP180 NC16A IgG were found to associate with the severity of the classical phenotype of the disease [35,36,37,38], we found that the erosion/blister BPDAI score was higher among patients with C3 deposition, but without reaching the level of statistical significance. The latter may stem from the relatively small number of patients in whom this score was available, rendering the study statistically underpowered to disclose significant differences.

Patients with C3 deposition in the current study demonstrated less prominent neutrophilic infiltrates in lesional skin specimens. Intriguingly, a similar finding was found in the study of Romeijn et al. [17] where no association was established between complement deposition in DIF and blister cellularity. These findings are somewhat surprising, given that experimental models of BP have demonstrated that complement activation by anti-BP180 IgG resulted in neutrophil accumulation and cleavage of proteins of the DEJ including BP180 by neutrophil elastase, thus contributing to subepidermal blister formation [10,39]. The presence of neutrophils in the skin without C3 deposition might be interpreted by the observation that binding of anti-BP180 IgG in cultured human keratinocytes induces the secretion of interleukin (IL)-8, which may lead to neutrophil attraction in a complement-independent manner [40]. This hypothesis is substantiated by previous observations in C4-deficient neonatal mice, that, although entirely resistant to the pathogenic effect of anti-BP180 IgG, developed clinical blisters after injection of IL-8 or neutrophils in the skin [10]. Another explanation may lie in the direct activation of local inflammatory cells by Fcγ receptors [41].

It should be noted that 15.9% of our study population evolved BP without complement deposition, signifying that complement-independent mechanisms may mediate the development of BP in this subgroup. Our study suggests that although highly important and influencing the immunological and histological features of BP, epidermal BMZ complement deposition is not indispensable for the development of BP.

Patients with isolated complement deposition presented with lower levels of anti-BP180 NC16A and anti-BP230 IgG as well as with lower seropositivity of anti-DEJ autoantibodies as compared to patients with complement and additional immunoreactants deposition. Likewise, the detection of multiple immunoreactants by DIF was associated with a more severe clinical phenotype in systemic lupus erythematosus, and it probably reflects the recruitment of a more intense inflammatory response [42].

### 4.3. Strengths and Limitations

The current study was based on a relatively large and well-characterized cohort of patients with BP to clarify the clinical implication of a widely employed diagnostic assay. Although the role of complement has been vigorously investigated in experimental models, it is yet to be firmly established in a clinical setting. The main limitation of our study arises from the retrospective collection of data that was originally retrieved for fulfilling diagnostic criteria rather than for research purposes. However, the routine workup in our department is extensive, thus enabling us to profile the investigated patients. Selection bias could not be thoroughly excluded owing to the tertiary-care referral center setting that may render the study susceptible to overlooking mild cases managed by outpatient dermatologists. It is noteworthy that only one antibody against complement (C3) was utilized, and that the use of different antibodies might increase the external validity of the findings. The histopathological analysis was based on a systematic review of the histopathological reports by two researchers rather than on reviewing the skin specimens.

In conclusion, the current study depicted that deposition of complement is observed in 84.1% of DIF analyses in BP. The presence of C3 deposition was significantly associated with the seropositivity of anti-BP180 NC16A autoantibodies and the absence of neutrophils in lesional histological specimens. Relative to patients with deposition of C3 alongside other immunoreactants, those with isolated C3 deposition had lower levels of anti-BP180 and anti-BP230 IgG autoantibodies, and lower frequency of eosinophil-dominant inflammatory infiltrate. The current study reinforces the view that complement is an important factor in the pathogenesis of BP. However, a sizable proportion of patients developed BP on a complement-negative skin, indicating that other complement-independent mechanism might be implicated in the disease.

## Figures and Tables

**Figure 1 jcm-09-03928-f001:**
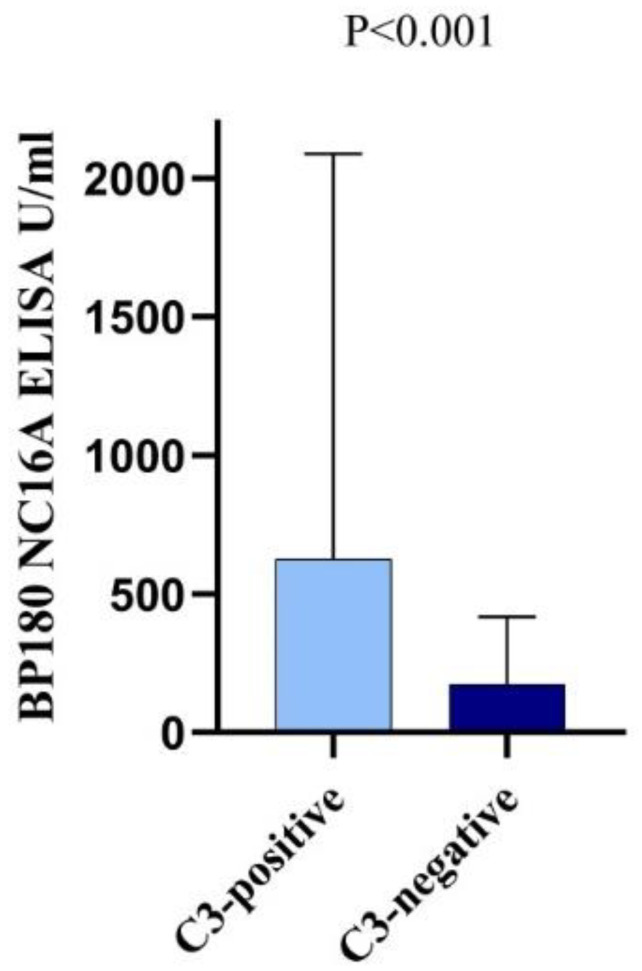
Anti-BP180 NC16A IgG levels of BP patients with linear C3 deposition relative to BP patients without C3 deposition at the dermal–epidermal junction by direct immunofluorescence microscopy. The lines represent the mean values.

**Figure 2 jcm-09-03928-f002:**
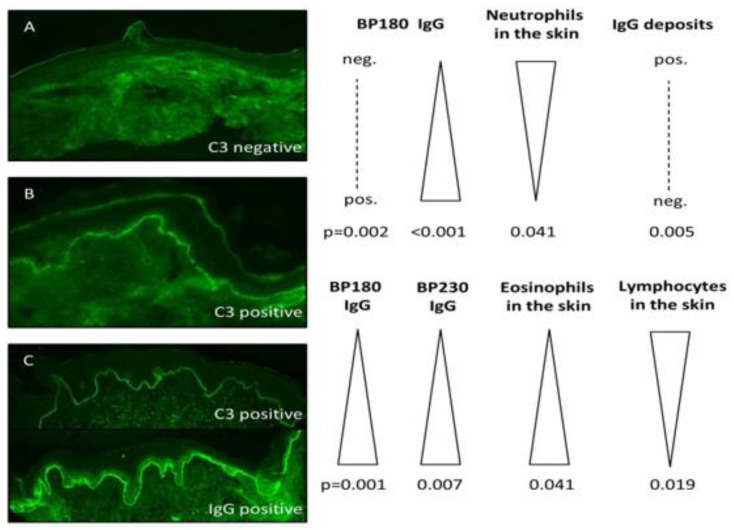
Visual abstract summarizing the main study findings. The figure includes three illustrative images of direct immunofluorescence microscopy from perilesional skin biopsies of BP patients. The immunoserological and histological features were compared between three groups. The comparison between (**A**) and (**B**) represents a comparison between patients without and with C3 deposition by direct immunofluorescence, respectively. The comparison between (**B**) and (**C**) represents a comparison between patients with isolated C3 deposition and patients with C3 deposition alongside IgG/IgA, respectively.

**Table 1 jcm-09-03928-t001:** Demographic and clinical characteristics of BP patients with C3 deposition by DIF as compared to BP patients without C3 deposition.

	BP with C3 Deposition (*n* = 196)	BP without C3 Deposition (*n* = 37)	*p* Value
**Age at diagnosis; years**			
Mean (SD)	79.1 (10.1)	80.0 (7.7)	0.613
Median (range)	80.6 (49.6–98.2)	80.7 (64.4–96.6)	
**Sex, *n* (%)**			
Male	85 (43.4%)	16 (43.2%)	0.989
Female	111 (56.6%)	21 (56.8%)	
**Distribution of bullous lesions; *n* (%)**			
Head and neck	54 (27.6%)	9 (24.3%)	0.685
Trunk	146 (74.5%)	28 (75.7%)	0.879
Limbs	161 (82.1%)	35 (94.6%)	0.057
Hands/feet	83 (42.3%)	12 (32.4%)	0.260
Mucous membranes	25 (12.8%)	1 (2.7%)	0.075
**DPP4i-associated BP, *n* (%)**	17 (8.7%)	4 (10.8%)	0.677
**Mean BPDAI severity score (SD) ^a^**			
Erosion/blister cutaneous activity	22.2 (16.0)	18.4 (10.5)	0.341
Urticaria/Erythema activity	12.4 (15.3)	15.5 (17.2)	0.492
Pruritus score	18.5 (9.3)	21.8 (8.1)	0.152
Damage score	2.3 (3.4)	1.7 (2.2)	0.352

Abbreviations: BP, bullous pemphigoid; *n*, number; SD, standard deviation; DPP4i, dipeptidyl peptidase-4 inhibitors; BPDAI, bullous pemphigoid disease area index. ^a^ Was available in 101 patients. **Bold**: significant values.

**Table 2 jcm-09-03928-t002:** Immunological characteristics and laboratory findings of BP patients with C3 deposition by DIF as compared to BP patients without C3 deposition.

	BP with C3 Deposition (*n* = 196)	BP without C3 Deposition (*n* = 37)	*p* Value
**BP180 NC16A ELISA ^a^**			
Seropositivity, *n* (%)	164 (86.3%)	24 (64.9%)	**0.002**
ELISA value, mean (SD); U/mL	645.2 (1418.5)	172.5 (243.9)	**<0.001**
**BP230 ELISA ^b^**			
Seropositivity, *n* (%)	25 (48.1%)	7 (43.8%)	0.762
ELISA value, mean (SD); U/mL	79.7 (160.5)	181.8 (483.1)	0.418
**Indirect immunofluorescence seropositivity, *n* (%)**			
Salt split human skin ^c^	180 (93.8%)	34 (91.9%)	0.676
Monkey esophagus ^d^	148 (78.3%)	30 (81.1%)	0.706
**Linear deposits of immunoreactants by direct immunofluorescence, *n* (%)**			
IgG	152 (77.6%)	36 (97.3%)	**0.005**
IgA	18 (9.2%)	5 (13.5%)	0.418
IgM	6 (3.1%)	1 (2.7%)	0.907
Two immunoreactant deposition	134 (68.4%)	5 (13.5%)	**<0.001**
Three immunoreactant deposition	21 (10.7%)	0 (0.0%)	**0.037**
**Histological findings, *n* (%)**			
Cell-rich infiltrate ^e^	59 (80.8%)	8 (66.7%)	0.266
Dominance of eosinophils ^f^	89 (80.2%)	12 (66.7%)	0.197
Dominance of neutrophils ^f^	6 (5.4%)	1 (5.6%)	0.979
Dominance of lymphocytes ^f^	16 (14.4%)	5 (27.8%)	0.154
Presence of eosinophils ^g^	122 (93.1%)	19 (90.5%)	0.663
Presence of neutrophils ^g^	39 (29.8%)	11 (52.4%)	**0.041**
Presence of lymphocytes ^g^	84 (64.1%)	18 (85.7%)	0.051
**Eosinophil count, mean (SD); cells/µL ^h^**	1295.0 (1072.9)	1026.3 (749.8)	0.909
**C-reactive protein, mean (SD); mg/L ^i^**	29.2 (26.7)	32.2 (38.4)	0.736

Abbreviations: BP, bullous pemphigoid; *n*, number; SD, standard deviation; CI, confidence interval; ELISA, enzyme-linked immunosorbent assay. ^a^ Was available in in 227 patients. ^b^ Was available in in 68 patients. ^c^ Was available in in 229 patients. ^d^ Was available in in 226 patients. ^e^ Was available in 85 patients. ^f^ Was available in 129 patients. ^g^ Was available in 152 patients. ^h^ Was available in 122 patients. ^i^ Was available in 89 patients. Anti-BP180 NC16A and anti-BP230 antibody levels were measured via enzyme-linked immunosorbent assay; cut-off: 20.0 U/mL. **Bold**: significant values.

**Table 3 jcm-09-03928-t003:** Factors associated with C3 deposition by DIF as identified by logistic regression model.

	Univariate OR	95% Confidence Interval	*p* Value	Multivariate OR	95% Confidence Interval	*p* Value
**Age at diagnosis ≥80.6 years**	0.95	0.47–1.91	0.880			
**Male sex**	1.01	0.50–2.04	0.989			
**Head and neck involvement**	1.18	0.52–2.67	0.685			
**Trunk involvement**	0.94	0.42–2.12	0.685			
**Limbs involvement**	0.26	0.06–1.14	0.057	0.87	0.17–4.52	0.873
**Hand and feet involvement**	1.53	0.73–3.22	0.260			
**Mucosal involvement**	5.26	0.69–40.11	0.075	2.26	0.27–19.3	0.455
**BP180 NC16A ELISA seropositivity**	**3.42**	**1.55–7.54**	**0.002**	**4.25**	**1.38–13.05**	**0.012**
**BP230 ELISA** **seropositivity**	1.19	0.39–3.68	0.762			
**Positive indirect immunofluorescence**						
Salt-split human skin	1.32	0.36–4.94	0.676			
Monkey esophagus	0.84	0.35–2.06	0.706			
**Linear deposits of immunoreactants in direct immunofluorescence**						
IgG	**0.10**	**0.01–0.72**	**0.005**	0.20	0.02–1.84	0.156
IgA	0.65	0.22–1.87	0.418			
IgM	1.14	0.13–9.73	0.907			
**Histology**						
Cell-rich infiltrate	2.11	0.55–8.00	0.266			
Presence of eosinophils	1.43	0.29–7.11	0.663			
Presence of neutrophils	**0.39**	**0.15–0.98**	**0.041**	**0.33**	**0.12–0.91**	**0.033**
Presence of lymphocytes	0.30	0.08–1.06	0.051	0.32	0.08–1.23	0.098
**Peripheral eosinophilia**	0.69	0.14–3.29	0.641			

Abbreviations: OR, odds ratio; CI, confidence interval; ELISA, enzyme-linked immunosorbent assay. Only variables demonstrating *p* < 0.200 in the univariate analysis were subject to inclusion in the multivariate logistic regression model. **Bold**: significant values.

**Table 4 jcm-09-03928-t004:** Demographic and clinical characteristics of BP patients with isolated C3 deposition by DIF as compared to BP patients with C3 deposition in conjunction with labeling of IgG and/or IgA.

	BP with Isolated C3 Deposition (*n* = 41)	BP with C3 Alongside Other Immunoreactants Deposition (*n* = 155)	*p* Value
**Age at diagnosis; years**			
Mean (SD)	78.0 (11.6)	79.4 (9.7)	0.437
**Sex, *n* (%)**			
Male	16 (39.0%)	69 (44.5%)	0.528
Female	25 (61.0%)	86 (55.5%)	
**DPP4i-associated BP, *n* (%)**	3 (7.3%)	14 (9.0%)	0.677
**Mean BPDAI severity score (SD) ^a^**			
Erosion/blister cutaneous activity	17.7 (16.0)	18.4 (10.5)	0.173
Urticaria/Erythema activity	12.4 (14.9)	23.2 (16.1)	0.185
Pruritus score	16.3 (9.8)	19.1 (9.2)	0.255
Damage score	2.5 (3.5)	2.3 (3.4)	0.860
**BP180 NC16A ELISA ^b^**			
Seropositivity, *n* (%)	30 (78.9%)	134 (88.2%)	0.140
ELISA value, mean (SD); U/mL	273.6 (388.3)	738.1 (1561.5)	**0.001**
**BP230 ELISA ^c^**			
Seropositivity, *n* (%)	3 (30.0%)	22 (52.4%)	0.203
ELISA value, mean (SD); U/mL	16.7 (20.0)	94.7 (175.4)	**0.007**
**Indirect immunofluorescence seropositivity, *n* (%)**			
Salt split human skin ^d^	34 (85.0%)	146 (96.1%)	**0.010**
Monkey esophagus ^e^	28 (70.0%)	120 (80.5%)	0.151
**Histological findings in lesional skin specimens, *n* (%)**			
Cell-rich infiltrate ^f^	7 (87.5%)	52 (80.0%)	0.611
Dominance of eosinophils ^g^	12 (63.2%)	77 (83.7%)	**0.041**
Dominance of neutrophils ^g^	1 (5.3%)	5 (5.4%)	0.976
Dominance of lymphocytes ^g^	6 (31.6%)	10 (10.9%)	**0.019**
Presence of eosinophils ^h^	19 (90.5%)	103 (93.6%)	0.600
Presence of neutrophils ^h^	6 (28.6%)	33 (30.0%)	0.896
Presence of lymphocytes ^h^	14 (66.7%)	70 (63.6%)	0.791
**Eosinophil count, mean (SD); cells/µL ^i^**	1038.9 (886.6)	1386.1 (1122.9)	0.150
**C-reactive protein, mean (SD); mg/L ^j^**	27.2 (26.5)	29.7 (26.9)	0.753

Abbreviations: BP, bullous pemphigoid; *n*, number; SD, standard deviation; CI, confidence interval; ELISA, enzyme-linked immunosorbent assay. ^a^ Was available in in 101 patients. ^b^ Was available in in 190 patients. ^c^ Was available in in 52 patients. ^d^ Was available in in 192 patients. ^e^ Was available in 189 patients. ^f^ Was available in 73 patients. ^g^ Was available in 111 patients. ^h^ Was available in 131 patients. ^i^ Was available in 103 patients. ^j^ Was available in 77 patients. Anti-BP180 NC16A and anti-BP230 antibody levels were measured via enzyme-linked immunosorbent assay; cut-off: 20.0 U/mL. **Bold**: significant values.

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
