# Peer review of "Presence of Cutaneous Complement Deposition Distinguishes between Immunological and Histological Features of Bullous Pemphigoid—Insights from a Retrospective Cohort Study"

_jcm, 2020, doi:10.3390/jcm9123928_

Round 1

Reviewer 1 Report

Ständer et al. present here a large retrospective study demonstrating that C3 deposition in DIF microscopy occurs among 84% of patients with BP, suggesting a role of complement activation in the pathogenesis of BP in human in line with data generated in mouse models. The  clinical, immunological and histological features described here may have important translational implications as many therapeutics targeting the complement pathway are nowadays in clinical development. 

However, a weakness of the study and as mentionned honestly by the authors,  is the use of a single anti-C3 antibody. Antibody clone, isotype and staining method should be described. To strengthen the analysis, a validation of the antibody should be provided with the use of an isotype control. 

There are some minor points which require to be corretcted:

  • Conclusion claimed 83.9% instead of 84.1% claimed in the abstract, suppl figure and in the discussion.
  • line 224, a typo "C5A7C5aR1" for C5a/C5aR1
  • line220, a typo "NC16A.."
  • line 240, a typo "autoantibodiesin BP"

Author Response

However, a weakness of the study and as mentionned honestly by the authors,  is the use of a single anti-C3 antibody. Antibody clone, isotype and staining method should be described. To strengthen the analysis, a validation of the antibody should be provided with the use of an isotype control.

Reply: We highly appreciate this important query. Fluorescein-conjugated goat polyclonal IgG to human complement C3 was utilized as a reagent in immunofluorescence assay and tissue staining. The total protein was measured using the Biuret procedure with bovine albumin as the standard. F/P ratio was calculated using absorbance at 491.5 nm. Please note highlighted changes under “2.2. Definition of covariates”.

This antibody was validated before its introduction and demonstrated higher sensitivity and specificity rates. In addition, the autoimmune laboratory in our institution is internationally certified

There are some minor points which require to be corretcted:

    Conclusion claimed 83.9% instead of 84.1% claimed in the abstract, suppl figure and in the discussion.

    line 224, a typo "C5A7C5aR1" for C5a/C5aR1

    line220, a typo "NC16A.."

    line 240, a typo "autoantibodiesin BP"

Reply: We highly appreciate the reviewer`s awareness and apologize for the typos. Please note highlighted corrected changes.

Reviewer 2 Report

The Authors present a very interesting study about the presence of complement in DIF microscopy in BP patients. The aim of this study was to investigate if complement deposition was associated with distinct morphological, immunological, and histological characteristics in BP patients.

Methods are sound and the conclusions are adequate.

However, I one comment: The Authors should specific the titre of BP180-NC16A antibodies in the group of patients with only C3 deposition at DIF. As, a low titre of BP180-NC16A antibodies could undermine the diagnosis of BP.

Author Response

One comment: The Authors should specific the titre of BP180-NC16A antibodies in the group of patients with only C3 deposition at DIF. As, a low titre of BP180-NC16A antibodies could undermine the diagnosis of BP?

Reply: We thank the reviewer for the comment.

Table 4 clearly specifies the titers and seropositivity rate of anti-BP180 NC16A among patients with isolated C3 deposition as compared to those with deposition of C3 in conjunction with other immunoreactants. We found that patients in the former subgroup had lower mean levels of anti-BP180 NC16A levels.

Reviewer 3 Report

This reviewer has several comments, questions and criticisms:

  1. Study population/diagnosis: had all the 3 criteria be fulfilled for the inclusion of a given patient into the study?
  2. Patients included: were these 233 different individuals or were certain patients recorded repeatedly?? It is quite likely that the DIF pattern of a given patient changes during the course of the disease.
  3. The predominating infiltrating cell type in BP lesional skin is the eosinophil. It should be stated more clearly whether C3 deposition along the BMZ of perilesional skin correlated with the eosinophil counts in the lesion and peripheral blood and, importantly, with the severity of the disease. Figure 2 points in this direction but doesn't include clinical data,

Author Response

  1. Study population/diagnosis: had all the 3 criteria be fulfilled for the inclusion of a given patient into the study?

Reply: We highly appreciate this important question. Yes, patients were required to fulfill all the aforementioned criteria to gain eligibility for the study.

  1. Patients included: were these 233 different individuals or were certain patients recorded repeatedly?? It is quite likely that the DIF pattern of a given patient changes during the course of the disease..

Reply: We highly appreciate this important comment. The analysis was established on the DIF deposition pattern that the patients demonstrated at their presentation and before the administration of any immunosuppressive treatment. As the reviewer kindly specified, the deposition pattern in DIF may change throughout follow-up, which may be caused, among other factors, by long-term immunosuppressive treatment.

  1. The predominating infiltrating cell type in BP lesional skin is the eosinophil. It should be stated more clearly whether C3 deposition along the BMZ of perilesional skin correlated with the eosinophil counts in the lesion and peripheral blood and, importantly, with the severity of the disease. Figure 2 points in this direction but doesn't include clinical data,?

Reply: We thank the reviewer for the important comments. Aligning with the reviewer`s recommendation, Table 2 and Table 4 demonstrates the association of (i) C3 deposition and (ii) C3 isolated deposition with eosinophilic inflammatory infiltrate in lesional skin specimens as well as with peripheral eosinophilia. Moreover, Table 1 illustrates the association of C3 deposition with all components of the BPDAI severity score as the reviewer requested.